# Vulnerable Road Users and Connected Autonomous Vehicles Interaction: A Survey

**DOI:** 10.3390/s22124614

**Published:** 2022-06-18

**Authors:** Angélica Reyes-Muñoz, Juan Guerrero-Ibáñez

**Affiliations:** 1Computer Architecture Department, Polytechnic University of Catalonia, 08860 Barcelona, Spain; 2Faculty of Telematics, University of Colima, Colima 28040, Mexico; antonio_guerrero@ucol.mx

**Keywords:** connected vehicles, pedestrians, automated vehicles, machine learning, deep learning

## Abstract

There is a group of users within the vehicular traffic ecosystem known as Vulnerable Road Users (VRUs). VRUs include pedestrians, cyclists, motorcyclists, among others. On the other hand, connected autonomous vehicles (CAVs) are a set of technologies that combines, on the one hand, communication technologies to stay always ubiquitous connected, and on the other hand, automated technologies to assist or replace the human driver during the driving process. Autonomous vehicles are being visualized as a viable alternative to solve road accidents providing a general safe environment for all the users on the road specifically to the most vulnerable. One of the problems facing autonomous vehicles is to generate mechanisms that facilitate their integration not only within the mobility environment, but also into the road society in a safe and efficient way. In this paper, we analyze and discuss how this integration can take place, reviewing the work that has been developed in recent years in each of the stages of the vehicle-human interaction, analyzing the challenges of vulnerable users and proposing solutions that contribute to solving these challenges.

## 1. Introduction

According to the World Bank, since 2020 more than 56% of the world’s population live in urban areas [1]. This agglomeration of people in urban areas causes serious mobility problems. The World Health Organization mentions that the number of vehicles circulating in big cities has increased uncontrollably, which means risks of more accidents, especially for Vulnerable Road Users (VRUs), including pedestrians, cyclists, and motorcyclists, among others.

Over the last decade, the automotive industry, along with research and development groups, have focused on creating intelligent vehicles with self-driving capabilities, known as connected autonomous vehicles (CAVs), which aim to increase the safety of passengers, road users and at the same time to contribute to reduce traffic accidents, road congestion, environmental pollution levels, etc. [2,3]. CAVs can detect and classify objects that are close to them and can notify the driver and other road users about the situation. For example, pedestrian detection state (pedestrians with intention to cross, pedestrians that stops suddenly or start running), detection of a traffic signal, detection of objects on the road, among others. CAVs can take real time control of certain operations with the aim of avoiding accidents.

CAVs should interact with all the elements that make up the ecosystem, including VRUs [4]. However, beyond the technical challenges related to automate driving, the success or failure of this type of CAVs is closely related to their acceptance and social integration within the vehicle traffic ecosystem.

In this review, we focus on analyzing the whole process of interaction between VRUs and CAVs. First, we examine the operating principles of a connected autonomous vehicle and explain the concept of VRUs. Secondly, we describe the technologies involved in the VRU-CAVs interaction process, describing them from two categories: sensing technologies, and algorithms that provide the intelligence to the CAVs. Thirdly, we analyze all the stages involved for a CAV to interact with VRUs, we make an in-deep literature review of the different papers that have been published for each of the interaction stages. Finally, we close this work showing the existing challenges in the VRU-CAVs interaction and the conclusions of the paper.

## 2. Vehicular Traffic Ecosystem

The road traffic ecosystem is seen as the entire travel environment on streets and roads that is used by vehicles and all kind of road users to move from one point to another. The vehicular traffic ecosystem is composed of several elements that must interact with each other to maintain a safe, accident-free environment. The elements that make up a vehicular traffic ecosystem are vehicles, VRUs (elderly pedestrians, children, disability people, cyclists, motorcyclists, and lately light commuting vehicles such as scooters, skateboards, electric scooters), traffic infrastructure (traffic signals, traffic lights, streets roads, etc.). The ecosystem also includes communications infrastructure (cellular networks such as 4G, 5G, wireless networks such as WiFi6, Bluetooth, and emerging networks such as Sigfox, among others). However, traffic ecosystem has changed in recent years and there are new elements such as CAVs (with different levels of automation), sensing infrastructure and automated electric vehicles. The traffic ecosystem focuses on increasing safety, improving traffic flow conditions, and reducing pollution levels. Figure 1 shows different vehicles that provide mobility services into a new vehicular traffic ecosystem.

### 2.1. Connected Autonomous Vehicles (CAVs)

In recent decades, the vision of automotive manufacturers is focused on the creation of intelligent vehicles that, on the one hand, offers all the mobility capabilities offered by currently vehicles, and on the other hand, have capabilities that allow them to perceive and understand the driving environment in which they are circulating, being able to perform the driving task with minimal or no human intervention. Precedence research released a report where it mentions that the autonomous vehicles market was 94.43 billion in 2021 and is projected to be around 1808.44 billion by 2030 [5].

The concept of CAVs refers to vehicles that are equipped with intelligent driving assistance systems and telecommunications technologies to establish communication with elements of the driving environment. CAVs are classified based on levels of automation, which were defined by the Society of Automotive Engineers in 2014 [6]. These levels of automation are based on the degree of human interaction during the driving task. The levels range from 0, where the driver has full control of the driving task, to 5 where the vehicle, through its implemented automation systems, can control all driving tasks dynamically without the intervention of a human driver. Figure 2 shows the different levels of automation.

A CAV is controlled by a set of heterogeneous autonomous driving systems, which are made up of several components that contribute to perform specific functions within the driving process. A functional and technical architecture of an autonomous vehicles was presented in [8], explaining from the technical point of view the integration of hardware and software inside the vehicle, and from the functional point of view, showing the processing blocks of all the activities performed by the vehicle to work correctly and efficiently. The functional architecture proposed in [8] consists of five main blocks: perception, planning and decisions, vehicle motion and control, system supervision and data exchange and communication control.Perception block. The function of the perception block is to create a representative model of the world surrounding the vehicle through the data received, both by the sensors installed in the vehicle, as well as external data generated by other elements of the ecosystem (pedestrian wearable networks, road side units, infrastructure, data processed in cloud services or fog). It also uses static data from the environment (such as digital maps, rules, routes) or environmental conditions (weather conditions and exact position in real time). The main perception tasks are object detection, localization, and object tracking. Localization integrates data from different sources, such as LiDAR (Light Detection and Ranging), Global Positioning System and Inertial Measurement Unit to increase the accuracy of the result. The implementation of particle filters are widely used for localization systems and have been shown to achieve accuracy levels of up to 10 cm [9,10,11,12]. Object detection consists of identifying and classifying the different objects, through the application of intelligent algorithms, which are detected through the set of sensors implemented in the CAVs. Trajectory tracking consists of identifying and predicting the possible path that an object will follow when it is in motion to avoid a risky situation.Planning and decision block. The purpose of this block is to generate the navigation plan for the vehicle, with the representative model of the world created within the perception block, and data information such as destination point, traffic rules, and maps, among others. This system must make a series of decisions to generate a safe and efficient real-time action plan. Its three main tasks are prediction, route planning and obstacle avoidance. Prediction is related to the function that the vehicle must perform to ensure that it can move safely within the driving environment [13]. Route planning focuses on defining the path to be followed by the vehicle within a dynamic traffic environment. To generate the movement plan, there are several context factors such as the state of the vehicle (speed, direction of movement, geo-reference, etc.), information from the vehicle’s travel environment (dynamic and static obstacles, driving spaces, etc.) and traffic regulations. Context factors help to create a safe travel path searching for all possible paths and filter them to select the best movement alternative. However, this type of evaluation and discrimination demands a large number of computational resources, which could affect the response time of the navigation plan. Generally, solutions are based on trajectory optimization through computationally intensive algorithms, trying to find a balance between optimization and computational time [14,15]. Obstacle avoidance refers to avoiding a collision situation with other elements located within the driving environment that endanger the safety of people. Through productive actions based on traffic predictions, measurements of minimum distances or time to collision with the object are used by the obstacle avoidance systems to make appropriate decisions and re-plan the navigation route of the vehicle. Reactive actions can make use of radar sensor data to avoid the detected obstacles.Motion and vehicle control. This block is in charge of the execution of the trajectory generated in the previous block through motion commands that control actuators inside the vehicle.System supervision. This block oversees checking the correct operation of hardware and software components of the vehicle to maintain the safety of all road users. It is based on the ISO (International Organization for Standardization) 26,262 functional safety standard [16], which is an adaptation of the IEC (International Electrotechnical Commission) 61,508 standard [17].Data exchange and communication control. This block is responsible for managing the entire data exchange process with the other elements of the road traffic ecosystem. All information travels over the network using one or more radio interfaces.


Figure 3 presents a functional architecture for CAVs [8].

The architecture of Figure 3 has a data source called external data, which is integrated to enrich the data set to be used in the CAVs perception process. The dataset comes from the process of collecting data that is shared by other vehicles, VRUs through their personal wearable networks, and traffic infrastructure (surveillance cameras, sensors installed on streets and roads). Data will extend the vision of the CAVs being able to identify hidden objects located within their vision coverage (e.g., a pedestrian who is hidden by another object such as a vehicle or a pedestrian that is approaching the intersection at a corner where walls do not allow the pedestrian detection (the perception system of the CAVs).

In connected autonomous vehicle, the data exchange and communication control functionality are integrated into the functional architecture. This block coordinates the communication and the transmission of emergency and notification messages that are sent from the vehicles on board unit to the rest of vehicles, pedestrians and the road side units. The vehicle will always be connected to the network using different radio interfaces. There is an architecture that details how notification messages are broadcasted through platforms such as 4G, WiMAX, etc. [18]. In addition, it is required to develop adaptive quality of service routing schemes that can quickly redirect traffic and alert notifications when the established routes are no longer available. In [19] there is a general review about which protocols, techniques and technologies would fit best for CAVs applications. The study details technical aspects of Transmission, Quality of Service, Security, Location and there is an in-depth analysis of the routing aspect, specifically focusing on which protocols are the best option to communicate vehicles with different Road Side Units (RSUs). Authors implemented a sensor technology and made different tests to analyze bandwidth limitation, delays, etc. In this context, packet delivery ratio, bit error rate, delay and connection duration have been analyzed in [20]. Some authors have discussed architectural issues and wireless technologies that support inter-vehicular communication, discussing outstanding challenges for enabling the deployment and adoption of inter-vehicular communication technology and how to combine these technologies in a cooperative way to exploit the advantages and cover the limitations of each of them [21,22,23,24].

CAVs will implement intelligent algorithms to select the data exchange interfaces. The selection process involves different parameters such as the type of application to be used (mission critical application, entertainment application, driver assistance application, among others), the requirements of the application (bandwidth, minimum delay, percentage of lost packets) and the level of quality of service that the network technology can offer [19].

### 2.2. Vulnerable Road Users (VRUs)

Within the areas of transport and road safety, the term vulnerable group has been used to refer to a specific section of road users such as walkers and cyclists. According to Ptak [25], the first time a similar term was used was in the 1950s, when it referred to unprotected road users, who were later called VRUs. The term VRUs has become very relevant in recent years in the transport and road safety environment. In 2013, the World Health Organization used the term VRUs to include pedestrians, cyclists, and motorcyclists. The United States Department of Transportation National Strategy on Highway Safety defined the term VRUs as “road users who are most at risk for serious injury or fatality when they are involved in a motor-vehicle-related collision. These include pedestrians of all ages, types and abilities, particularly elderly children and people with disabilities, bicyclists and motorcyclists” [26]. For the European Union’s Intelligent Transportation System Directive [27], the term VRUs is specified as “non-motorized road users, such as pedestrians and cyclists as well as motor-cyclists and persons with disabilities or reduced mobility and orientation”.

The Organisation for Economic Co-operation and Development proposed the creation of new categories according to their mobility and their ability to manage within the road environment including all users who have minimal protection when circulating in vehicular traffic areas and therefore can easily be injured or even killed in an environment dominated mainly by vehicles [28].

In this survey, we classify VRUs in the following categories (Figure 4):
Distracted road users. They are the type of pedestrians walking in the road traffic ecosystem, who are distracted by some extra activity they are doing. Gen-erally, the activity they are doing may be using a cell phone, conversing with an-other person, or thinking about something else.Road users inside the vehicle. We refer to passengers of a CAVs or drivers of a traditional vehicle. People into the car could be elderly or sick people who could suffer an eventuality while traveling. Passengers/drivers can be continuously monitored through Body Sensor Networks or through monitoring devices such as cameras or sensors that are implemented in the steering wheel. These sensors al-ways verify the driver’s conditions and can detect risk situations (fatigue, stress, distraction, among others). On the other hand, passengers in a CAVs can also be monitored through sensors implemented in the seats to detect physiological changes that lead to risky situations.Kjh.Special road users. This category refers to people who have a very low travel speed, including elderly and children. They are the most at risk within a road en-vironment. Around half of accident pedestrian occurs at sites remote from cross-ing facilities, with many occurring when parked vehicles obscured driver vision. Children appear suddenly to cross the road while being masked by stationary ve-hicles, failing to look properly, or being careless. Elderly tends to move slowly and are more likely to be less able to judge the path and speed of vehicle.Users of transport devices. In recent years, there has been a trend to decrease the usage of cars and use lighter modes of transportation, especially for the last mile. For this reason, this category refers to users in a transport device who are not pro-tected by and external mechanism, such as skates, scooters, roller skis or skates as well as by kick sleds or kick sleds equipped with wheels.Animals. They are all types of animals that could be within the road driving zone, such as cats, dogs, horses, among others.Road users with disabilities. These are the type of pedestrians moving through the road traffic ecosystem but who have a disability (such as blind people, deaf people, people in wheelchairs or people with assistive devices such as canes, crutches, etc.).


## 3. CAVs and VRUs Interaction

For CAVs to be a success, they will need to be always in direct communication with the different elements of the road traffic ecosystem. The interaction between the CAVs and the VRUs is of great importance since poor or deficient communication can have fatal consequences. Communication is of great relevance because on the one hand, if the vehicle knows the intentions of the VRUs, then the vehicle could react and avoid a collision with the VRUs that could cause severe damage to the VRUs, up to and including loss of life. On the other hand, if the VRUs knows the intentions of the CAVs, then the VRUs may react positively and more confidently to be able to, for example, cross the road.

The traditional process of interaction between the pedestrian and the driver of a vehicle is carried out through non-verbal communication, including facial gestures, eye contact, hand signals and even sounds [29,30]. This informal language indicates the actions to be taken by the vehicle (stop and give way to the pedestrian, continue driving, etc.), and the actions to be taken by the pedestrian (stop, cross the street) to avoid a possible eventuality [31]. However, with the addition of CAVs to the streets, the entire interaction process will change as automation levels advance [32].

The vehicular traffic ecosystem will start to become a hybrid environment where traditional vehicles (non-automated and non-connected), semi-autonomous vehicles, CAVs, and VRUs will coexist. Therefore, non-verbal communication will no longer work in all interactions. Some researchers believe that CAVs will no longer need such nonverbal communication [33]; however, some other authors believe that the lack of nonverbal communication will lead to distrust and rejection of CAVs by pedestrians.

The base of the CAVs-VRU interaction is to predict what other users (cyclists, scooters, pedestrians, motorcyclists, drivers, among others) intend to do next in order to make a proper movement decision. Connected and automated vehicles will not only detect objects, but also predict the behavior of other users and notify their intention to the rest of road users.

In [34] authors explain the interaction without non-verbal communication occurs through two stages. The first stage is called communication of awareness and describes the entire process that must be carried out for the CAVs to detect and identify the VRUs. The second stage, called communication of intent, describes the capabilities of the CAVs to notify the VRUs of its next action (stopping or not stopping for the pedestrian). In this article, we consider that a third stage is necessary, called broadcast communication, which describes the different types of communication between the CAVs and the RSUs, infrastructures, VRUs and other CAVs.

### 3.1. CAVs-VRU Interaction Process

The CAVs-VRU interaction process is made up of different stages. Figure 5 shows in a general way the stages that must be executed for a successful interaction process.

Object detection. This function is a prerequisite for the CAVs to be able to perform autonomous navigation. The CAVs detects all types of objects within its driving environment. Based on the number of objects, it will set a guideline to calculate its possible future trajectory. For this it will make use of a set of sensors (such as LiDAR, cameras, radar, and Global Positioning System) that will allow it to detect objects, their position, their distance and keep track of objects (moving and stationaries) [35].

Object classification. Object classification is the phase that allows the vehicle to identify each type of object being detected as the vehicle moves through the road traffic ecosystem (such as pedestrians, traffic lights, road signs, walkways, and much more). In addition to classifying them you need to know the exact distance between itself and each object around it.

Intention prediction. This stage refers to predicting the behavior of the VRUs so that the CAVs can redesign its trajectory and actions to prevent accidents. Intent identification of a VRUs allows to classify its activities to predict, for example, whether a pedestrian will cross the street or stop for a vehicle to pass [36].

Trajectory and tracking. Trajectory prediction is one of the essential components to increase VRUs safety. Through this prediction the CAVs estimates the future state of each of the moving elements within the road driving environment. The objective is basically to anticipate the next action through the analysis of previous actions. This will increase safety for VRUs.

Communication. Communication is focused on how the CAVs will let the VRUs know of its intent to move. In the absence of such non-verbal communication with the driver, alternatives need to be sought for the exchange of information between the CAVs and the VRUs.

### 3.2. Technologies for Interaction between CAVs-VRUs

The CAVs-VRU interaction process involves several technologies such as a driving assistance system to reduce the risk of accidents and to reduce the percentage of human error [37].

#### 3.2.1. Sensing Technology

For CAVs-VRU interaction, advanced sensors are used to detect movement and thus reduce the risk of accidents. The heterogeneous sensing mechanisms are integrated into the On board Unit of the vehicles to generate a robust data acquisition system that is interconnected through different communication media such as Local Interconnect Network (LIN), Controller Area Network (CAN), Media Oriented Systems Transport (MOST), Low-Voltage Differential Signalling (LVDS), Ethernet, among others. Each link used for interconnection has different characteristics. Local Interconnect Network (LIN) is a unidirectional bus that has a transmission capacity of 20 Kbps and is used to connect sensors and actuators to Electronic Control Units (ECUs). LIN uses a single cable connection and the maximum transmission distance between two ECUs is 40 m.

CAN is a bus based on message protocols for the interconnection of controllers and devices in order to establish communication between them. CAN buses are classified into high-speed, which achieves communication speeds of up to 1 Mbps, and low-speed fault-tolerant with speeds of up to 125 kbps and transmission distance up to 40 m. Flexible Data bus is a variant that can be transmitted at different data rates by varying the message size. It achieves transmission speeds 8 times faster than traditional CAN.

MOST (Media Oriented Systems Transport) is a standard for high-speed interconnection of multimedia components in vehicles. MOST uses a ring topology, performing one-way transfer within the ring and transmits data via light pulses. It uses synchronous data transmission to exchange audio, video and data signals via optical fiber or electrical conductor. MOST provides a data-rate of 25 up to 150 Mbps using optical fiber in a shared ring topology.

Automotive Ethernet is a bus used to transport a large amount of data in real time with very low latency. Automotive Ethernet uses a point-to-point network technology and defines the 100Base-T1 standard to achieve transmission speeds of 100 Mbps. Currently, a new task force, called 802.3cy, is working on the development of the automotive PHY layer standard for 25, 50 and 100 Gbps.

Low-Voltage Differential Signalling (LVDS) is a transmission system based on inexpensive media, such as twisted pair that transmits signals at high speeds. This standard specifies only the physical layer. It is used for high-speed video, graphics, and video camera data transfer. Its speeds of 655 Mbps have made it a viable alternative for connecting self-driving vehicle camera systems.

Gigabit Multimedia Serial Link (GMSL) serializer and deserializers are high-speed communication interfaces that support high bandwidth requirements, complex interconnections, and data integrity that are applied in ADAS and infotainment systems. It uses a point-to-point connection with support for 4K video. In general, it operates through a serializer on the transmitter side to convert the data to a serial stream and implements a deserializer on the receiver side to convert the serialized data to a word format for processing, reaching transmission speeds of up to 6 Gbps. You have transmission distances of around 15 m.

Table 1 shows a summary of the different communication media used in cars for the interconnection of their devices.

Sensors are interconnected to small electronic control units that control each of the different functions that the CAVs must execute to perform the self-driving process. One of the key points of the interconnection of the different elements of the CAVs is its topology. In-vehicle network architecture models there has been a migration from an architecture model based on a central multi-bus gateway to a functional domain controller architecture model (Figure 6) that performs all functions with more few ECUs [38]. In this architectural model, the ECUs are relocated within a functional domain, and a series of updates are applied to them to adapt them to the new vehicle features. However, in recent years, a new architectural model is being worked on, where ECUs are viewed as generic computing units used to perform functions that demand high processing requirements (such as object detection and classification, object intention and trajectory prediction, among others zonal ECUs are used to perform traditional ECU functions according to vehicle characteristics.

For the vehicle to interact with the other elements of the environment it needs to “see” everything around it. That ability of the vehicle allows it to detect and recognize all the elements within its driving environment (other vehicles, traffic signs, VRUs, to name a few) [39]. A series of sensors installed inside and outside the CAVs are used to collect all the information from its environment. These sensors are used in a complementary manner to increase the accuracy of object recognition [40]. All the information collected by the sensors is analyzed to construct the route that the vehicle will use to move from point A to point B, and thus send a series of instructions to the vehicle’s control systems (braking system, acceleration system, steering system). According to a report presented by the company YOLE Développement (Villeurbanne, France), there are three types of sensors that dominate the autonomous vehicle market: LiDAR, image sensors and RADAR sensors [41].

LiDAR is a type of sensor that works through a sonar, using laser light pulses to recreate a map of all objects near the vehicle. The basic architecture of a LiDAR system consists of four components: the transmitter that emits laser pulses, the receiver that receives the bounced light pulses, the optical analysis system whose function is to process the input data, and a computer to display a live three-dimensional image of the system environment. The computer measures the time it takes for the light pulse to travel back and forth, and with this value calculates the distance that the light pulse traveled, and also the angle of the LiDAR unit and the firing angle of the light pulse. To avoid failures due movement and changes of angles, it is necessary to integrate the inertial measurement unit. By integrating data from this unit with collected data, it is possible to have the tracking of thousands of points per second, allowing building the digital image of the environment.

The millions of points received by LiDAR form a concept called “point cloud”. This information is processed at different stages. There is a stage called Clustering which causes multiple “point clouds” to be overlaid to give the objects a recognizable shape. Subsequently, the classification stage performs the identification of each type of objects, and they are classified into categories (such as pedestrians, cars, traffic signs, etc.). Finally, the modelling stage assigns predictive contexts to each of the scanned objects to map all possible movements.

One of its key features is its depth perception accuracy, knowing how far away an object is from a few centimetres to 60 m away. LiDAR is used by CAVs to generate a 3D detailed picture of the area through a point cloud [42], which allows them to have a better knowledge of the distance of objects and is not affected by textured or textureless reflective surfaces. LiDAR application area within self-driving vehicles focuses on obstacle detection, road users and lane markers [43,44,45].

Some advantages of the Lidar sensor are: (i) Speed and accuracy to collect data, (ii) Active illumination sensors improve efficiency because it is not affected by light variations (e.g., day and night), (iii) Not affected by geometric distortions, (iv) Data collection is not affected by extreme weather conditions (such as extreme sunlight).

As any type of sensor, LiDAR has a series of limitations in its operation, among which we can mention (i) its high cost of operation, (ii) in specific weather situations such as rainfall, snowfall or low hanging clouds the sensor is affected in its performance due to the refraction effect, (iii) when generating a huge amount of data, it is necessary a great processing capacity to analyze the data.

RADAR sensors detect objects of interest and estimates some features such as distance, size, location, motion, relative velocity of an object with respect to the transmitter [46]. Its operation is based on the principle of reflection, whereby a series of radio waves are transmitted through space until they collide with an object and are reflected back to the transmitter. With this information, the details of the object can be calculated. RADAR sensors operate at different frequencies (24, 74, 77 and 79 GHz), which allows them to work at different ranges [47]. Ranges used in RADAR have different functions:Short-range radars are used in functions such as blind-spot monitoring, lane-keep assistance, and parking assistance.Medium-range radars are implemented for obstacle detection functions within the range of 100 to 150 m and the beam angle varies between 30° and 160°.Long-range radars are used for automatic distance control and brake assistance.


The advantages of RADAR sensors are: (i) their robustness of operation even in unfavorable conditions such as snow, clouds, fog, allowing them to collect data from the environment without being affected in its performance [48], (ii) they provide the exact distance of an object due to the use of electromagnetism, (iii) they allows to calculate the speed of displacement of a moving object, which complements the data of the object’s position and its possible trajectory, (iv) they have the ability to simultaneously target several objects, since its radio signals operate over a wide area, which allows it to collect data from several objects simultaneously, (v) since different return angles of the signals can be received, a 3D image of the environment can be generated, (vi) their cost is lower compared to other sensors.

The disadvantages of this type of sensors are: (i) Time to target an object is not so efficient, due to the time it takes for the signals to reach the object and return to the transmitter, (ii) the range or coverage of this type of sensors is shorter (200 feet) compared to other sensors such as LiDAR, (iii) they can suffer interference from other signals traveling through space, altering the data transmitted, (iv) they cannot identify the type or the shape of the object correctly, (v) these sensors can only detect objects that are within their line of sight, if an object is hidden by another object, the sensor cannot detect it and therefore will not be able to react quickly.

Vision sensors (cameras) facilitates that automated vehicles detect pedestrians, objects and read traffic signs [49]. Cameras will scan the road, processing information about what they see and responding to an obstacle in its path. To process the information, cameras will have a software architecture that combines conventional image-processing algorithms with AI-driven methods and embedded it on a high-performance system-on-chip with an integrated microprocessor.

Vehicles will have a system of cameras covering all viewing angles to provide a 360° panoramic view of the external environment, facilitating the detection of VRUs and surrounding traffic conditions.

Cameras are classified as visible or infrared. The former can be monocular [50] or stereo [51]. Monocular cameras use a single camera to create a series of images, but they cannot capture depth information. However, using dual-pixel autofocus hardware and image processing algorithms, depth can be calculated in the image [52,53]. In autonomous vehicles, two such cameras are usually installed to create a binocular camera system. Stereo cameras are more similar to the depth perception behavior of the animal eye. These cameras use two image sensors separated by a suitable distance known as the baseline. The authors in [54] mention in their study that the baseline used by autonomous vehicles is 75 mm. The disparity produced by the two cameras allows calculating the depth within the image. Cameras capture wavelengths between 380–780 nm, which is the wavelength range of visible light. Ordinary camera sensor chip perceives areas that are invisible to the human eye. The active night vision cameras are sensitive to near infrared (800–1000 nm). Then when the filter technology is used to filter out the visible light, what the camera sees should be an image composed of infrared radiation [55]. Those cameras are less susceptible to variations in illumination or drastic changes in illumination [56]. Higher range, resolution and field of view pose many challenges to overcome with new electronic device innovations. Automotive sensor integration technology would require to have video processing units integrated in the on board unit of the vehicle (it requires a deep analysis of bandwidth resources, delay, jitter, etc.) or to transmit large amounts of high-resolution digital video data over single data lines such as GMSL. The advantages of this type of sensors are: (i) easy distinction of shapes, and faster identification of the type of object based on the information collected, resembling the capacity of a human eye, (ii) high resolution for the detection of objects, (iii) its cost is not high compared to LiDAR, (iv) when complemented with infrared illumination it has a better performance in night driving.

According to the strengths and limitations of each of the sensors, the authors in [57] made a comparative analysis of the sensors used in autonomous vehicles. Table 2 presents the summary of the different characteristics of the most used sensors in the automated vehicles.

Each type of sensor has its strengths and weaknesses, and sensor fusion is used to obtain more accurate results in automated driving systems. Sensor fusion is the process of taking the data collected by different types of sensors to better interpret the environment around the vehicle. Through the data from each sensor, sophisticated algorithms, known as fusion algorithms, determine more precisely the position of each of the objects located within its driving zone. The fusion algorithms use a prediction equation and an update equation to estimate the kinematic state of the objects.

The equations use two models for their calculations. The motion model is focused on the motion dynamics of the object, while the second model, known as the measurement model, focuses on the dynamics of the sensors implemented in the vehicle. By applying the two equations, the exact position information of each object is obtained.

The prediction equation integrates data from previous predictions and the motion model calculates the current state of the vehicle. The update equation combines data from the sensors and the measurement model in order to update the prediction state. Thus, at the end of the process, a range of possible state values is available.

In [58], Cotra provides two equations that describe a motion model that represents the knowledge about the dynamics of the object. The motion model uses a deterministic function f() and a random variable q_k__−__1_. Thus, the state x_k_ is a function of the previous state x_k__−__1_ and a random motion noise q_k__−__1_, which is stochastic (Equation (1)). The measurement model is formulated as a deterministic model that receives the current state x_k_ as well as a random variable representing the measurement noise of the sensor type r_k_ (Equation (2)). Combining the two models yields a density called posterior distribution over the state, and a region of values for x_k_ can be described from all observed values. The prediction (Equation (3)) and update (Equation (4)) equations are computed to express density.
x_k_ = f (x_k−1_, q_k−1_)(1)
y_k_ = h (x_k_, r_k_)(2)
p (x_k_|y_1:k−1_) = ∫ p (x_k_|x_k−1_)p (x_k−1_|y_1:k−1_)dx_k−1_(3)
P (x_k_|y_1:k_) = p (y_k_|x_k_)p (x_k_|y_1:k−1_)/p (y_k_|y_1:k−1_)(4)

In [59], the authors explain that there are different modalities in which sensor fusion can be performed. In High Level Fusion each sensor performs object detection, tracking functions and finally fusion. This type of fusion is used because of its low complexity; however, it may present inadequate information due to object overlapping. Mid-level fusion integrates multi-target features (such as color, location, among others) that are obtained from each of the sensors and with these data performs the recognition and classification process of the fused features. Finally, low-level fusion integrates data from each sensor type at the lowest level of abstraction, which improves the accuracy of object detection.

In [60], authors present a series of the most commonly used algorithms for sensor fusion, classifying them into four categories: (i) based on the central limit theorem, (ii) Kalmar filters, (iii) based on Bayesian networks and (iv) based on convolutional neural networks.

Algorithms based on the central limit theorem focus their operation on the argument that as the sample size of any measurement increases, the average value will tend to a normal distribution. Thus, as more samples are obtained from the sensors, an average value of the set will be obtained, and therefore less noise will be present in the sensor fusion algorithms [61,62]. Kalman filter uses input data from different sensors and estimates unknown values without being seriously affected by high levels of noise in the signal. This type of algorithm is applied in the process of pedestrian detection and trajectory prediction, basing its operation on a series of predictions and state updates [63,64,65]. Bayesian networks are applied in the update equation used in sensor fusion, which integrates the measurement and motion models. Bayesian networks are applied in real-time navigation processes in advanced driver assistance systems [66,67]. Deep learning-based algorithms perform the processing of raw data from the different sensors and in this way extract the features that allow it to perform intelligent driving tasks such as pedestrian detection [68,69,70].

#### 3.2.2. Software Technology

Artificial Intelligence (AI) has become the core for the development of self-driving systems. AI refers to the effort to replicate or simulate human intelligence in machines so that they can perform tasks that are so far only performed by humans (such as visual perception, speech recognition and decision making). Through AI, machines learn based on the experience they acquire, adjust themselves according to that learning and can thus perform tasks similar to humans. AI automates learning by making use of data, performing a deep analysis of data, and making accuracy.

In the self-driving environment, different technologies such as machine learning, deep learning, and computer vision are commonly applied.

##### Machine Learning and Deep Learning

Machine learning (ML) is being applied in advanced driver assistance systems (ADAS) such as (i) object detection, (ii) object identification and classification, (iii) object localization and motion prediction. ML is divided into three categories: supervised learning, unsupervised learning, and deep learning. Supervised learning is based on the use of labeled data used for knowledge generation and within which the results of the operation are previously known. By means of these results, the model learns and adjusts so that it can adapt to new data that are introduced to the system. Unsupervised learning makes use of unlabeled data whose structure is not known. The objective is to obtain important information of which the reference of the output variables is not known. Finally, reinforced learning builds models that increase performance using each of the results of the interactions.

For the different types of ML to be executed, a series of algorithms need to be implemented. ML algorithms can be classified into four categories: regression algorithms, pattern recognition, cluster algorithms and decision matrix algorithms. Table 3 shows the characteristics and uses of each of the algorithm categories.

Deep Learning (DL) is a branch of Machine Learning that is based on a multi-layered model that is used for feature extraction as well as for representation learning at various levels of abstraction [82]. DL makes use of a concept called Artificial Neural Networks (ANN). ANN is a series of learning algorithms that are based on the functioning of the human brain to learn a huge amount of data. Within ANN, the primary element taken as a basis is known as the neuron, which represents the fundamental unit of the DL model [83]. The interconnection of these neurons to form a processing layer is called perceptron [84]. Its basic operation consists of performing a task repeatedly with the objective of improving the result. For this purpose, it uses “deep layers” for progressive learning to take place.

The overall operation of DL is composed of two stages known as training and inference. The training phase focuses on performing labelling on a large amount of data to determine the adaptive properties. On the other hand, the inference phase oversees labelling new unseen data, making use of previously acquired knowledge. This method helps the complex vehicle perception tasks to be performed with the highest accuracy. In addition, DL is also known as deep structured learning as it consists of a set of interconnected layers, where the output of one layer is used as input to the next layer and by using nonlinear processing performs the feature extraction process.

ML uses a much smaller amount of data, while DL uses a huge amount of data to acquire the best result but demands a high-performance of the Central Processing Unit [85] (Figure 7).

##### Computer Vision

Computer vision is a subfield of ML focused on implementing the ability to “see” in machines to understand the surrounding environment. Using data acquisition systems such as cameras and sensors, a set of tools process and analyze images of the real world, which contributes to the automation of the driving process. They make use of artificial intelligence algorithms to decode the images to help them recognize shapes, figures, and patterns in the images.

One of the applications of computer vision in self-driving systems is object detection. This process consists of two steps: classification and localization. On the one hand, classification is performed by training with Convolutional Neural Networks (CNN) to recognize and classify objects. On the other hand, localization is applied by using non-max suppression algorithms [86], which selects the best bounding box based on the intersection over the union of the bounding boxes, omitting the rest. This process is repeated until the boxes cannot longer be reduced.

##### Augmented Reality

AR is a powerful tool that can be easily applied to the CAVs-VRU interaction offering ubiquitous situation awareness support. The first step is to determine what kind of information has to be made artificial or augmented. Second step in the situation awareness support will be the design of interfaces to help VRUs to understand the CAVs context information such as the dynamics from approaching vehicles or systems to navigate traffic situations. The AR information could be available with a multitude of mobile pervasive and context-aware applications. For example, (a) the pedestrian could be presented with safety corridors related to which vehicles will stop for them, (b) road specific alerts that a vehicle is approaching, (c) highlighting hazards considering blind people, elderly, children, etc., (d) CAV’s interfaces to predict the influx of pedestrians at schools, metro, etc. in order to find alternative routes, (e) VRUs could have information not just about the CAV’s intention to stop but also about where it intends to stop, (e) and in addition, there are some proposals [88] such as an augmented traffic light in the form of a virtual fence to stop pedestrians from crossing a vehicle lane.

AR hybrid approaches should be proposed to consider different smartphones, wearables, glasses, and user devices. Even, the possibility that some VRUs will not use any kind of device, so the information should be also available in the road site units, infrastructures or from the vehicle itself.

## 4. Stages for CAVs-VRU Interaction

Within the literature, different works can be found that contribute to solve each of the scenarios or stages of the interaction between CAVs-VRU.

### 4.1. Object Detection and Classification

The first phase of the CAVs-VRU interaction process is object detection and classification. For the CAVs, this is a primary task as it helps it to identify everything around it. The idea of this task is that the vehicle perceives everything around it in a very similar way to what the human eye does when the driver performs the driving task [89]. This will allow the intelligent control systems implemented in the CAVs to learn and take action [90].

One of the problems faced by object detection within the autonomous driving model is the high demand for processing large amounts of data, which places high performance requirements on the algorithms [91].

For obtaining object features, the algorithms used within the autonomous driving environment have been classified into two categories: (i) Machine learning algorithms using artificial features and (ii) deep learning algorithms based on features by convolutional neural networks.

Machine learning algorithms focus on feature extraction and classifiers [92]. For feature extraction, techniques such as Histogram of Oriented Gradients [93,94,95,96,97,98,99,100], Local Binary Pattern [101,102,103,104,105,106,107], Deformable Part Model [108,109,110,111,112,113], and Aggregate Channel Feature (ACF) [114,115,116,117,118] are included. On the other hand, methods such as Support Vector Machine (SVM) [94,105,119,120,121,122], Decision Tree [123,124,125,126], Random Forest (RF) [127,128,129,130,131,132] and Ada-Boost [81,119,133,134] are used for the classification process.

Within deep learning techniques, one of the best performance algorithms for feature extraction is CNN.

CNN is a DL architecture that has proven to have excellent results when applied to image classification, resulting in classification rates of up to 100% accuracy [135]. CNN’s operation can be explained as follows. Successive perceptrons learn complex features in a supervised manner by propagating classification errors. Finally, the last layer represents the category of the output images [136,137]. Recall that being DL-based, no prior training module is used, but everything is carried out implicitly through supervised training, avoiding manual feature extraction [137].

Deep Learning based algorithms for VRUs detection are divided into two categories: (i) region proposal algorithms, (ii) regression-based algorithms. Region-based algorithms focus their operation on two processes. First, it generates candidate regions where it is expected to contain the object to be detected, this is accomplished by means of region recommendation algorithms. Subsequently, applying the CNN, the final detection box is obtained. In this category, one of the most widely used networks for VRUs detection is known as Region-based CNN (R-CNN). R-CNN combines regions and CNN features. It has two stages: (i) the first one identifies a number of regions that tentatively could contain the object to be identified (they are called region proposals) and (ii) classify the object in each proposed region. In [138], the authors apply CNNs to detect pedestrians in situations of dark environments or illumination variation, showing that by using multi-region features they obtain better results in detection accuracy. In [139], authors proposed an algorithm with R-CNN and applied it to pedestrian detection. Their results proved that the region proposals generated by their method are better than the selective search. Fast R-CNN is a variant of R-CNN, but the main difference is that it takes as input the complete image and a set of proposed objects and thus produces a feature map. Subsequently for each object proposal, a layer containing a set of regions of interest generates a feature vector of the feature map. Within the literature, there are several works that apply R-CNN for the detection of VRUs [140,141]. Faster R-CNN is an enhancement of Fast R-CNN that adds a region proposal network with the objective of generating region proposals directly and not using an external algorithm. This results in a faster generation of region proposals, which better fit the data [142]. Several works where Faster R-CNN is applied focus on solving the problem of occlusion of VRUs detection in natural scenarios [143,144,145], small object detection [144,146]. Authors in [147] presented a solution where they apply a variant of CNN, known as Mask Region-based CNN and instance segmentation was used to detect pedestrians crossing the streets, showing results of over 97% accuracy in the detection process.

Regression-based algorithms do not use the concept of region. Instead, the input image is only processed once, and both the category and the target border can be regressed on multiple image positions [143]. The most representative algorithms in this category are YOLO (You Only Look Once) [141], SSD (Single-Shot MultiBox Detector) [148] and RetinaNet. The authors of [149] proposed a method where they apply a CNN to the whole image, dividing it into multiple regions, improving the speed of detection, a feature of utmost importance to avoid risky situations [150]. This work underwent some improvements by applying YOLO2 [151] and YOLO3 [152], providing a balance between speed and accuracy in the detection of VRUs. Authors in [153] propose a loss function to improve sample classification during the training process in order to solve the problem of sample imbalance, which generates a poor detection result. YOLO version 4 was proposed in 2020 [154] as an efficient alternative for pedestrian detection due to its advances in accuracy and real-time processing performance [130].

### 4.2. Intention Prediction

The intention prediction is related to the actions that the VRUs will take in a short period. For example, when a pedestrian wants to cross a street, some signals it provides are key to identify its intention, such as turning its head to be able to identify if any vehicle is approaching the crossing. They, upon identifying that a vehicle is approaching the crossing, stop and do not cross. This identification of intentions is a key element in the CAVs-VRU interaction. Some work is focusing on the use of past and current information for intention prediction by VRUs through different neural network architectures [155]. Some works have proposed a dataset based on pedestrian trajectory data generated from the driver’s point of view to study the pedestrian intention prediction field [156,157].

In [158], the authors mention that methods for predicting pedestrian intention fall into two groups: (i) methods that approach the problem as a trajectory prediction issue with the objective of creating a route and identifying whether that route will cross the street [159,160,161] and (ii) methods that solve it as a binary classification problem that results in the pedestrian intention [162,163,164,165].

Those focused on trajectory prediction use neural networks to predict the individual trajectory, assuming a prior conversion from image coordinates to real world. They are generally applied to model interaction between people or between people and environment. Some proposed works incorporate scene information in the predictive models, taking into consideration that trajectories remain within the driving environment [166,167].

In [168], a model based on the clustering of the hidden states of all people within a neighbourhood is proposed. This work is improved in [169] by defining an attention mechanism that assigns a weight to each element participating within the driving environment based on its proximity. However, according to [158], this type of model suffers from several limitations such as the need for moving cameras to obtain a complete view of the scenario, which can cause errors in the accuracy of the trajectories. They do not make use of pedestrian pose information, which they say is a weakness because it is an important indicator of intent. In addition, by requiring multiple frames, it introduces significant delays in predictions.

Binary classification is the simplest method within the ML environment. It consists of categorizing the data points into two possible alternatives: for example, the pedestrian crosses the street or does not cross the street. These methods work based on two types of models. The first models use RGB inputs where they apply filters that either slide along the height and width (2D convolutions) or even add temporal depth (3D convolutions). Some papers using the 2D convolution model use Long Short-Term Memory (LSTMs) or feature aggregation over time to propagate information across time [170]. Other work uses 3D CNN and LSTM [164,171] to generate two feature vectors, based on a single pedestrian snippet input, which are concatenated for classification [172]. There are other methods that work directly with the skeleton of the individual [173,174] to reduce the amount of data (e.g., 17 joints of the skeleton compared to 2048 feature vectors), which results in a lower probability of readjustment.

### 4.3. Trajectory and Tracking

Trajectory prediction and tracking are two indispensable tasks within the CAVs-VRU interaction phase. Trajectory prediction targets where objects will be in the immediate future. This point is important because it can be noted that we do not have data collected by the sensors to corroborate the results obtained. What is used is past data to predict the future position as shown in [175,176]. On the other hand, object tracking focuses on knowing where the object is currently located. Therefore, this process makes use of sensor data that provide or support its current position.

In [177], the authors mention that there are two methods for trajectory prediction: (i) linear model and (ii) nonlinear model. Within the first method, the motion of the object cannot be accurately described [178]. Non-linear methods are based on data driven algorithms [179].

According to [180], data-driven methods using neural networks perform better than traditional methods. Some work adds the element of interaction between pedestrians within the driving environment, (making use of human-human interaction feature extraction [181,182], capturing the interactivity information between adjacent pedestrians [168]), to improve the trajectory prediction process. Other work applies inverse reinforcement to perform the pedestrian trajectory prediction process [183]. Other works such as [165,176,184,185] integrate different factors such as speed, location, direction of the pedestrian’s head and environmental into the process to predict intention and future trajectory.

Different variants of deep learning have been used for the trajectory prediction process. The three most used architectures are: (i) recurrent neural networks, (ii) convolutional neural networks and (iii) generative adversarial networks [186].

RNNs use a fully connected two-layer neural network within which the hidden layer implements a feedback loop, allowing sequential data to be modelled more efficiently. Some work uses the LSTM structure to learn pedestrian activity patterns and the environment within a scenario over a long-term period [187,188,189], the human body pose [190], and the influence of human [168].

CNN contains many convolutional, non-linearity, pooling, dropout, batch normalization, and fully connected layers. Based on the architecture used, the most significant discriminative information is included, which allows for a better level of precision in the identification of target objects. Authors in [191] proposed a model that uses a CNN fed with different types of information (such as historical trajectory, depth map, pose, and 2D-3D size information) that help it to better predict the trajectory of pedestrians. The problem of prediction of complex downtown scenarios with multiple road users is addressed by combining a Bayesian filtering technique for environment representation, and machine learning as long-term predictor [192]. Other work uses convolutional neural networks to predict average occupancy maps of walking humans even in environments where no human trajectory data are available [193].

Generative Adversarial Network (GAN) uses competing generator/discriminator architecture with the objective of reducing the fragmentation of conventional path prediction models and thus not computing costly appearance features. Compared to the other architectures it is more lightweight and is being used to achieve multimodality in trajectory prediction. Authors in [194] proposed a complete deep learning framework for multi-person localization and tracking. The proposed method uses GAN for human localization, which addresses the problems of occlusion and noisy detections by generating human-like trajectories with minimal fragmentation. Other works address the problem of the influence that pedestrians have on each other in determining the trajectory to be followed. In order to determine the trajectory authors apply concepts such as socially aware GAN, multimodal pedestrian behavior, scenario context, etc. [182,195]. GAN is also used to perform the prediction sampling process for any agent within the scenario [196]. In [197] there is a proposal that focus on possible failures and crashes of pedestrian trajectory with a prediction of several seconds. The research implements, though info-GAN a cost function to replace the L2 loss term.

### 4.4. Intention Communication Interfaces

Finally, the last phase of the interaction process is focused on how the pedestrian and the vehicle will exchange information about intentions or actions. Several studies have been conducted regarding the interaction between pedestrians and vehicles driven by a human [198,199,200]. However, as the level of automation increases in vehicles, it will be necessary to define mechanisms to replace the non-verbal communication currently used by pedestrians with drivers.

Previous work has focused on the interaction between autonomous vehicles and passengers [201,202,203]. However, as the objective is to generate a safe and efficient road driving ecosystem, it is necessary to generate mechanisms that cover the CAVs-VRU communication. The absence of a driver in an automated vehicle generates distrust in pedestrians because they cannot know the intentions of the self-driving vehicle [204,205].

The new traffic ecosystem will require elements that provide a fluid, natural communication that emulates the actual, intelligent interaction between the CAVs and the VRUs within their travel area, identifying their intentions and reacting to the actions they decide to take. In addition, it should provide the VRUs with relevant information such as its status and future behavior. With these two features in place, it will increase the safety and efficiency of the road traffic ecosystem and increase the trust and acceptance of CAVs.

In recent years, human-computer interfaces, known as eHMI (external Human-Machine Interface), which are placed on the outside of the vehicle, are being used as an alternative solution for relevant communication and dissemination (such as speed or intention of vehicle movement) to enable CAVs-VRU communication [206]. For the design of eHMI interfaces, the type of information to be communicated by the interface must be taken into consideration because it depends on several factors (such as simplicity, target audience, how we want the target audience to find out). In [197] author classifies the information in three categories: driving mode, intention and perception.

The driving mode is an important type of information, because as long as the vehicle is not fully automated, it is of great relevance to indicate to other users whether the vehicle is controlled by a person or by computers.

Intention is another relevant type of information since it will set the tone for the action that the pedestrian might execute. That is, as a vehicle and a pedestrian approach an intersection, it would be helpful for the vehicle to indicate to the pedestrian that it has already detected the pedestrian and that it will stop so that the pedestrian can cross the street safely. Finally, the perception of everything around the vehicle is important as a form of collaboration with other vehicles to avoid a risky situation for the pedestrian (for example, the non-detection of the pedestrian due to an obstacle that does not allow the sensors to detect it). There are still many doubts about the type of information that should be shared, but beyond that, the question that also arises is how the eHMI should be designed to communicate the information to the VRUs.

In [207], the authors classify interaction technologies into four categories: (i) visual, (ii) visual and acoustic, (iii) concepts with anthropomorphic elements and (iv) infrastructure external to the vehicle (Figure 8).

Visual Interfaces. This technology focuses on communication through interfaces that allow the display of information. The most used interfaces are screens (Figure 8a), LED strips (Figure 8b), holograms, or projections (Figure 8c). Displays are used to show messages through text or icons. Generally, the displays are placed at the front or rear of the vehicle; however, they can also be placed on the sides to cover the notification in all directions. The messages displayed can be of intent such as “cross” or “stop”, or more elaborate messages such as “after you”. In addition, you can make use of iconography to indicate that you have been detected and what intention the vehicle has (stop, do not stop, etc.) [208,209,210,211]. On the other hand, there are the LED strip interfaces, which are placed on the windshield or on the grill of the vehicle and work as a kind of traffic light to indicate to the VRUs the action to follow (stop, move forward, among others) [212]. Hologram or projection interfaces use lasers to project relevant information (using text messages or icons) onto the road surface.

Visual interfaces can also be used to communicate with hearing-impaired pedestrians. Using displays, the autonomous car could communicate with people through sign language [213,214]. 

Visual and Acoustic interfaces. These interfaces are simply an extension of the visual interfaces, including within their communication mode acoustic signals with the objective of extending the transmission of the message to people with visual impairments (Figure 8d). We must remember that this is not new, as this type of interface has been used for years in traffic light systems. In this way, although the vehicle will display the message on the screen, it will also include clear and concise verbal messages, for example “safe to cross” [215,216].

Anthropomorphic interfaces. This type of interfaces makes use of human characteristics to carry a communication that gives the VRUs a greater security to perform the actions. Specifically, efforts have been made to simulate eye contact with pedestrians (something that is currently used within the non-verbal language used in pedestrian-driver interaction). For this purpose, object detection and identification technologies and the use of an interface that shows the “eye movement” of the fake driver are integrated (Figure 8e). The idea of this interface is to be a more intuitive form of communication than the VRUs is used to [217,218]. Within this concept of interfaces, Jaguar Land Rover has generated a prototype, called “virtual eyes”, which aims to understand the level of acceptance that humans will have in relation to self-driving vehicles. The vehicle has implemented cartoon-like eyes, which are used to interact with humans in the road traffic ecosystem. Through eye contact with pedestrians, it notifies them that it is watching out for them [219].

From the VRUs’ point of view, efforts have been made to establish interaction directly from the pedestrian to the car. The cellular device is being used as a viable tool for pedestrian-to-car communication. One effort focuses on using the cell phone to share location data of both the car and the pedestrian through P2V (pedestrian-to-vehicle) communication mode [220,221]. The application performs calculations to establish the risk zone and thus reduce the likelihood of a collision. Other work focuses on the development of an ADAS applications for mobile devices that retrieves car and pedestrian location information to identify hazardous situations between VRUs and CAVs [222,223,224].

## 5. Challenges in the ACs-VRU Interaction

The idea behind the interaction between CAVs and VRUs is that technology will free pedestrians, similar to the autonomous vehicles free up the driver. However, it will be a long way before we have fully autonomous vehicles (level 5 driving automation [7]) introduced in the roads and urban environments. This new kind of environment will bring new challenges of user privacy, invasiveness, technology feasibility, inclusiveness, etc.

In the transition process of fully interconnected vehicles with infrastructures and road users it will be necessary that the whole mobility system should still adhere to the current rules and ways to communicate the vehicle’s intentions such as keeping speed, braking, or accelerating. A deep analysis of how human-driven vehicles will interact with CAVs will be needed and also how CAVs will influence walking behavior to reduce pedestrian fatalities.

CAVs should be designed to be understandable even for those VRUs who do not have the technology or have other limitations.

Standardization and training will be useful for all road users who may be able to distinguish automated vehicles from manually driven vehicles to learn to interact with them. In addition, contemporary urban design should consider unpredictable pedestrian behaviors by (a) braking to avoid striking pedestrians, (b) predicting trajectories of pedestrians, (c) providing early alerts to the road users about dangerous behaviors, (d) separating traffic by means of tunnels and bridges, among others.

The challenge in the interaction between VRUs and CAVs is not only for the VRUs to understand the actions that the CAVs will perform, but also for the CAVs to learn the communication language of some VRUs, such as cyclists. Cyclists use hand signals to indicate, for example, that they will make a left or right turn, whereupon the CAVs must slow down for the cyclist to perform the manoeuvre and avoid an accident. One of the solutions is to generate algorithms for the vehicle to learn the different signals generated by cyclists. At the same time, work should be carried out to raise awareness among cyclists to use this signalling code to have a correct communication with the CAVs.

Another challenge is in the training process of the algorithms, since the algorithm will be good if the amount of data is sufficient, and that all data are validated. However, the challenge is the data collection and labelling of the information. The collection can be solved by implementing collection systems within the driving support systems, currently implemented in many of the vehicles driving in urban areas, and that all this information is shared in centralized repositories to be used for the creation of data sets for algorithm training. In addition, these data sets can be enriched with data generated through simulation scenarios. On the other hand, data labelling is a crucial process for the optimal performance of algorithms responsible for self-driving tasks. One solution is the use of off-the-shelf labelling tool platforms that facilitate the data labelling and validation process.

Finally, it is necessary to consider different VRUs behavior between all countries where CAVs will operate. Furthermore, trust and acceptance are particularly challenging in countries that currently have very little advanced technology within the transport system [225].

## 6. Conclusions

The emergence and introduction of autonomous vehicles to the road traffic environment will generate a series of challenges that must be solved. One of them is the interaction between autonomous vehicles and the rest of the road users.

In this paper, we present a review of the interaction process between VRUs and autonomous vehicles. We analyze the road traffic ecosystem, identifying the evolution of the environment and the new elements that are being integrated. We describe the essential functions of an autonomous vehicle and define and describe the main categories and characteristics that make up the group of VRUs. Subsequently, we discuss from the technical aspect, the interaction process between VRUs and CAVs. The analysis revealed how learning technology is positioning itself as an essential element of the interaction process as it allows the autonomous vehicle to identify, classify and predict the behavior of VRUs, contributing to the reduction of the probability of a risky situation ending in fatal consequences for VRUs.

It is necessary to solve different challenges to improve the perception technologies and the definition of interfaces that facilitate communication and understanding of intentions by both VRUs and ACs. Although, many efforts have been made to address some of the challenges in the interaction between VRUs and ACs, there still are open problems pending, such as the improvement of the algorithms, training, dataset, among others, to increase the accuracy of all stages of the interaction process. The review also shows that eHMI interfaces are one of the driving forces that will facilitate the acceptance of CAVs. However, eHMIs still have not make the communication between VRUs and CAVs transparent and understandable.

CAVs-VRU interaction should be designed to guarantee the inclusion of the different requirements of all kinds of VRUs: children, elderly, people with disabilities, etc. The concurrent use of different safety modalities, each targeting a different human sense, seems a promising approach.

## Figures and Tables

**Figure 1 sensors-22-04614-f001:**
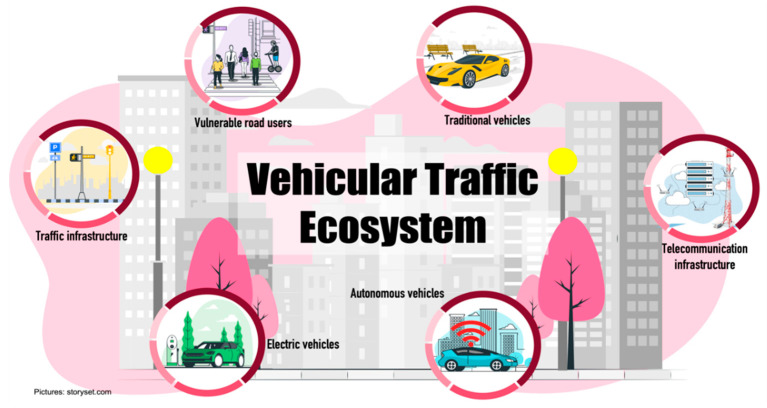
A new vision for the Vehicular Traffic Ecosystem.

**Figure 2 sensors-22-04614-f002:**
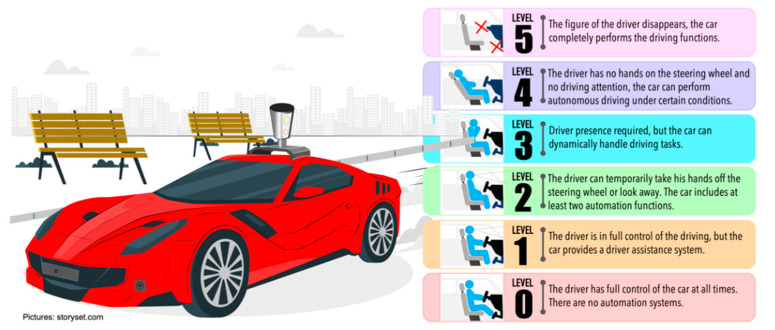
Brief description of the six levels of vehicle driving automation defined by the Society of Automotive Engineers. The figure is based on the content presented in [7].

**Figure 3 sensors-22-04614-f003:**
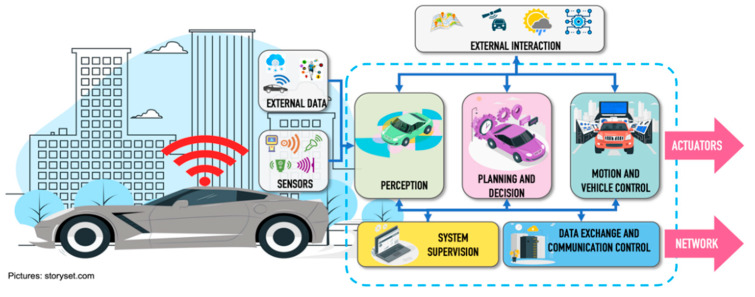
Representation of the functional architecture for a connected autonomous vehicle [8].

**Figure 4 sensors-22-04614-f004:**
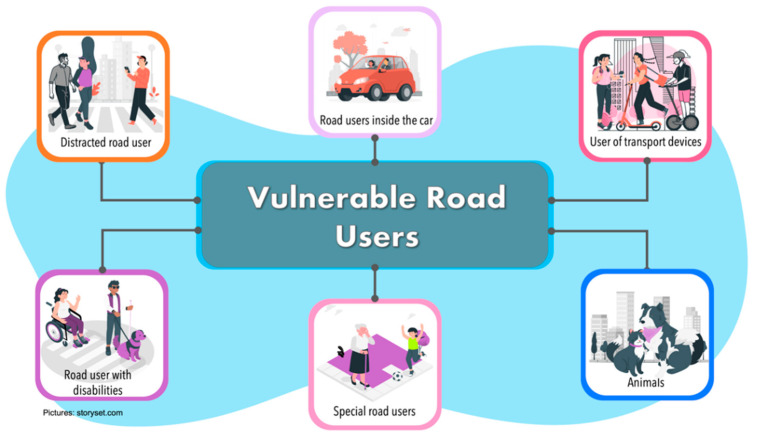
Representation of the different categories for VRUs.

**Figure 5 sensors-22-04614-f005:**
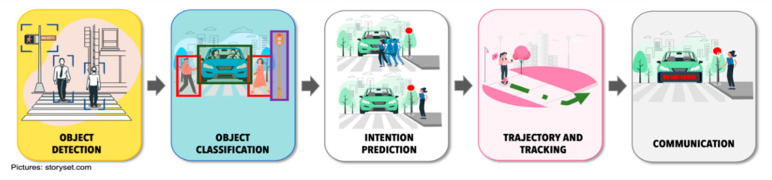
Representation of the stages of the CAVs-VRU interaction process.

**Figure 6 sensors-22-04614-f006:**
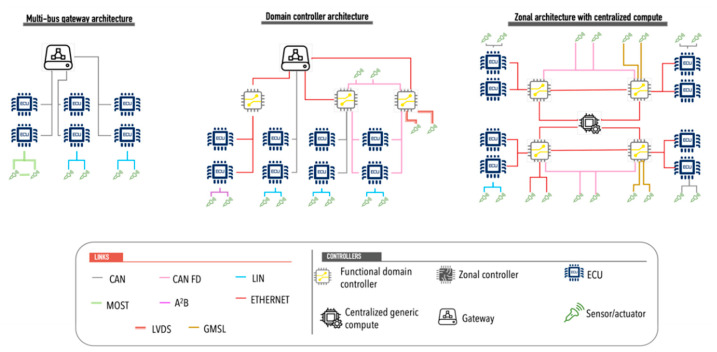
Representation of the different types of architectural models for ACs, adapted from [38].

**Figure 7 sensors-22-04614-f007:**
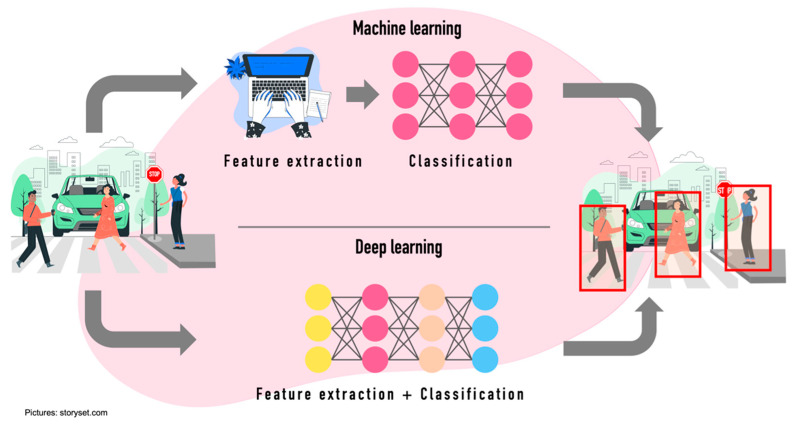
Representation of the difference between ML and DL. Based on [87].

**Figure 8 sensors-22-04614-f008:**
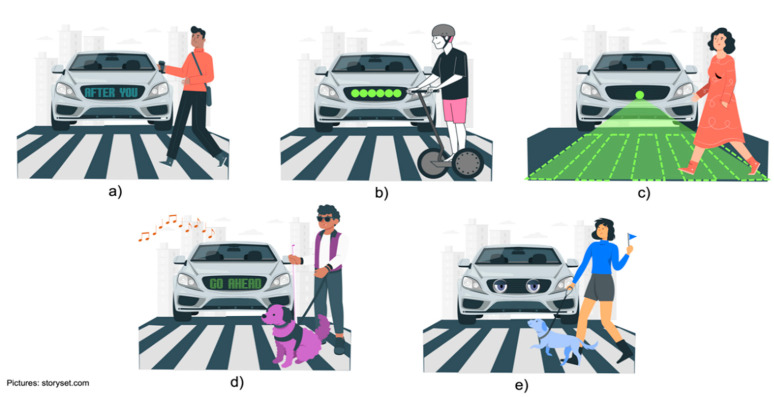
Representation of visual interfaces, (**a**) display on the front of the vehicle showing in text information on what the pedestrian should do, (**b**) LED strip lights on the front of the vehicle in different sequences of movement and colors according to the type of message, (**c**) projection of message on the road with visual elements to indicate to the pedestrian the option of “safe crossing”, (**d**) acoustic and visual interface to indicate the action to follow by the pedestrian, (**e**) vehicle with human appearance to emulate the communication by visual contact.

**Table 1 sensors-22-04614-t001:** Summary of autonomous vehicles’ communication media.

Media	Description	Transmission Speed	Usage	Distance
LIN	Single-wire unidirectional bus	20 kbps	This media connects sensors and actuators to ECUs. This media is used in applications such as cruise control, position sensor control, temperature control, sunroof, among others.	Up to 40 m
CAN	bus based on a message protocol	High speed up to 1 Mbps. Low speed 125 kbps	It is used for controller and device communication without the need for a computer host.	Up to 40 m
CAN FD	A variant of CAN that uses flexible data	8 Mbps	Used in communications with sensors at different transmission rates.	Up to 40 m
MOST	A standard used for interconnection of multimedia components that uses a ring topology, performing one-way transfer within the ring and transmits data via light pulses. Up to 64 devices can be connected to the network	From 25 up to 150 Mbps using optical fiber.	Used for audio and video applications in or out of the car. Most is the best transmission and multimedia control network most widely used in automotive electronics.	Up to 40 m
LVDS	Transmission system based on twisted pair, that transmits signals at high speeds.	655 Mbps.	A viable alternative for connecting self-driving vehicle camera systems.	15–20 m
GMSL	High-speed communication interfaces that support high bandwidth requirements, complex interconnections, and data integrity	Up to 6 Gbps	Used for ADAS and infotainment systems. It uses a point-to-point connection with support for 4 K video.	Using shielded twisted pair (STP) or coax cables of up to 15 m

**Table 2 sensors-22-04614-t002:** Summary of autonomous vehicles’ sensors features.

Feature	LiDAR	RADAR	Camera
Primary technology	Laser light pulse	Radio wave	Light
Range	∼200 m	∼250 m	∼200 m
Data rate	20–100 Mbps	0.1–15 Mbps	500 Mbps in high resolution
Resolution	Good	Average	Very good
Affected by weather conditions	Yes	Yes	Yes
Affected by lighting conditions	No	No	Yes
Detects speed	Good	Very good	Poor
Detects distance	Good	Very good	Poor

**Table 3 sensors-22-04614-t003:** Summary of Machine Learning algorithms categories.

Category	Usage	Description
Regression	This type of algorithm is used for autonomous vehicles for event prediction such as collisions, trajectory prediction.	The algorithms focus on establishing a method to define the relationship between a set of variables (which represent the characteristics) and a continuous target variable. Examples of such algorithms being applied in self-driving systems include Bayesian regression [71], neural network regression [72] and decision forest regression [73].
Patter recognition	This type of algorithm is used for CAVs for the object classification such as pedestrians, vehicles, cyclists, traffic signals.	This type of algorithm is used to perform data filtering to recognize instances of a category of objects by discarding irrelevant data points. They focus on reducing the data set through edge detection and fitting line segments and circular arcs to edges. These features are combined to define the object features to be recognized. The most applied recognition algorithms in Advanced Driver Assistance Systems (ADAS) are support vector machines (SVM) with histograms of oriented gradients [74] and principal component analysis (PCA) and Bayes’ decision rule [75] and k-nearest neighbor [76].
Cluster	This type of algorithm is implemented in autonomous vehicles for object classification and detection.	This type of algorithm groups data to discover its characteristics. It is generally used in situations with little data, with discontinuous data or with very low-resolution images. To solve this problem, it generates “center points” and a series of hierarchies that allow it to discover a series of common characteristics. Among the most used algorithms are K-Means [77], K-Medians [78] and Hierarchical clustering [79].
Decision matrix	The main use of this type of algorithms in autonomous vehicles is decision making.	The structure of this model focuses on a set of independently trained decision models, combining their predictions to generate the overall prediction, thus reducing the probability of errors in decision-making. Some examples of this type of algorithms are gradient boosting [80] and AdaBoosting [81].

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
