# Peer review of "Vulnerable Road Users and Connected Autonomous Vehicles Interaction: A Survey"

_sensors, 2022, doi:10.3390/s22124614_

Round 1

Reviewer 1 Report

1.      "CAVs can detect and classify objects that are close to it and can notify the driver about the situation, or through actuators" (line 35). List the specific situation.

2.      The statements "Figure 1 show the new vision of the vehicular traffic ecosystem (line 65) "and "offer all the mobility capabilities currently offered by vehicles (line 70) ". Please correct these sentences.

3.      Whether Road users inside the vehicle should be counted in the category of Vulnerable Road Users in Section 2.2? It should be known that passengers in fully autonomous vehicles do not interfere with the vehicle, vehicle and passengers should be regarded as the entire agent.

4.      The phrase "Gigabit Multimedia Serial Link (GMLS) (line 315)"is wrong and section 4 serial is repeated.

5.      Since deep learning is a branch of machine learning, can the two subsections 3.1.2.1 Machine learning and 3.1.2.2 Deep learning be merged?

6.      Are the interaction technologies in the "4.4.Intention communication interfaces" reasonable? And the authors do not give relevant literature descriptions, such as how deaf and blind people interact with CAVS?

7.      The authors list Telecommunications Technologies (4G, 5G, Bluetooth, WiFi, among others) (Line 59, Line77). Are there any different effects different effects on Vulnerable Road Users and Connected Autonomous Vehicles Interaction?

8.      Do children, the elderly, and the disabled have different effects on Vulnerable Road Users and Connected Autonomous Vehicles Interaction?

9.      Is there any basis for the statement CAV-VRU? Literature citations should not be written at the beginning of a sentence, such as literature"[56] [74] [75] [117] [158] [159] [181]…". The format of the literature [20][21][51] are incorrect. "Machine learning ML" should be written as "Machine learning (ML)".

10.   The format of the reference is incorrect, please revise it again.

Author Response

We thank the reviewer for his comments to improve our paper and we did our best to reach all his suggestions. In the following, we highlight how we addressed each specific concern of reviewer 1.

  1. "CAVs can detect and classify objects that are close to it and can notify the driver about the situation, or through actuators" (line 35-38). List the specific situation.

R = A sentence was added on line 35 to indicate situations where the CAV can notify the driver about some situations in the environment.

  1. The statements "Figure 1 show the new vision of the vehicular traffic ecosystem (line 65) "and "offer all the mobility capabilities currently offered by vehicles (line 70) ". Please correct these sentences.

R= The sentences were reformulated.

  1. Whether Road users inside the vehicle should be counted in the category of Vulnerable Road Users in Section 2.2? It should be known that passengers in fully autonomous vehicles do not interfere with the vehicle, vehicle and passengers should be regarded as the entire agent.

R = We reformulated point (ii) of section 2.2 to better explain why we consider people, who is traveling in the car, as part of the vulnerable group.

  1. The phrase "Gigabit Multimedia Serial Link (GMLS) (line 315)"is wrong and section 4 serial is repeated.

R = the spelling error was corrected, and the section numbering was updated.

  1. Since deep learning is a branch of machine learning, can the two subsections 3.1.2.1 Machine learning and 3.1.2.2 Deep learning be merged?

R = The machine learning and deep learning subsections were integrated into a single section.

  1. Are the interaction technologies in the "4.4. Intention communication interfaces" reasonable? And the authors do not give relevant literature descriptions, such as how deaf and blind people interact with CAVS?

R = Section 4.4 explains that our current human behavior should “evolve” or at least should be adaptable to be easily understood by autonomous vehicles. The new traffic ecosystem will require elements that emulate the current pedestrian-driver communication. A paragraph was added in section 4.4 to indicate how interaction with people with a hearing impairment can be performed. 

  1. The authors list Telecommunications Technologies (4G, 5G, Bluetooth, WiFi, among others) (Line 59, Line77). Are there any different effects different effects on Vulnerable Road Users and Connected Autonomous Vehicles Interaction?

R = The list of telecommunications technologies is the same for VRUs and CAVs. In order not to duplicate information, we eliminate one of the lists.

  1. Do children, the elderly, and the disabled have different effects on Vulnerable Road Users and Connected Autonomous Vehicles Interaction?

R = Yes, each group of users have specific requirements to be considered. The easies example is to compare children that may have unpredictable and fast movements with elderly that may require more time to cross the street. In addition, it is necessary to apply different algorithms for the identification of people in wheelchairs.

  1. Is there any basis for the statement CAV-VRU? Literature citations should not be written at the beginning of a sentence, such as literature"[56] [74] [75] [117] [158] [159] [181]…". The format of the literature [20][21][51] are incorrect. "Machine learning ML" should be written as "Machine learning (ML)".

R = The references [20][21][51] were corrected. In addition, the format of Machine learning ML was corrected by Machine learning (ML)

  1. The format of the reference is incorrect, please revise it again.

R = Format of all references was reviewed

Reviewer 2 Report

There are many flaws in the article, like inaccurate data, the lack of proper review for all the solutions, or poorly written sentences.  I have tried to present many of them below, but I consider the paper should be rewritten and maybe a more useful evaluation can be made when the information is properly structured and easy to be followed.

Figure 1: why autonomous vehicles appear twice? 

The correlation between Figure 3 and the text above should be better explained: why the actuator blocks are considered as "main', along with "System supervision" form the network section. Why, for example, Data exchange is not as important? To be more clear: you say in the text that there are 4 main blocks, but the Figure reveals more.

I strongly disagree with the statement in rows 161-163, that all the communications available are there to provide alternatives. This suggests that, if one communication mean is not available, one can simply use another and it works! But this is not true, every communication technique is designed for certain applications and are not always interchangeable. In this regard, the authors should present the existing communications, their applications and situations when one solution can be used instead of another. 

Figure 4 should present more accurate VRU categories. For example, you have "road user outside the vehicle", but this also includes "other road users", "special road user", even "user of transport devices", because all of them are outside a vehicle.

In chapter 3.2.1 are presented several communications technologies inside vehicles. A table that compare them would be useful, including main parameters and the applications.

Please, document some more the LASER technology and decide if it uses  pulsed laser, or laser beams. Or if it is the same thing and it doesn't matter.

Also, please document some more the video camera technology. First of all, you say that infrared camera can use wavelength down to 1nm. But this is not infrared, it's even beyond ultraviolet and towards X-rays! Second of all, combing the information presented, it results that video cameras can only work with GMLS. Are you sure this is the only solution? What about cameras with video processing units integrated. Do they require the same (huge) amount of communication bandwidth?

Please, present and detail the prediction equation and the update equation you are referring to in the text.

All the technologies presented in chapter 3.2 are from the vehicle point of view. Aren't any solutions from the VRUs point of view? Because interaction means "reciprocal action" and what you present do not suggest reciprocity.

In chapter 4 you introduced some solutions to interact with VRUs, but also from the vehicle point of view. There are more solutions out there, such as V2P, V2B and so on that you don't mention anything about.

In the end, I must emphasize again that the whole paper should be reconsidered and maybe a native English speaker should be involved. There are so many sentences that don't make sense, that is difficult for me to list them all.

And all the abbreviations should be explained in the text.

Author Response

There are many flaws in the article, like inaccurate data, the lack of proper review for all the solutions, or poorly written sentences.  I have tried to present many of them below, but I consider the paper should be rewritten and maybe a more useful evaluation can be made when the information is properly structured and easy to be followed.

R =We thank the referee for the careful review of our manuscript and we did our best to rewrite the paper as suggested. In the following, we highlight how we addressed each specific concern. 

  1. Figure 1: why autonomous vehicles appear twice?

R = Figure 1 was modified to correct the error of duplicity of autonomous vehicles. 

  1. The correlation between Figure 3 and the text above should be better explained: why the actuator blocks are considered as "main', along with "System supervision" form the network section. Why, for example, Data exchange is not as important? To be more clear: you say in the text that there are 4 main blocks, but the Figure reveals more.

R = We corrected the sentence to improve the correlation between figure 3 and the text. The number of blocks in the text was corrected to 5 instead of 4. Some paragraphs were reformulated and a new paragraph was added to describe the data exchange and communication control block.

  1. I strongly disagree with the statement in rows 161-163, that all the communications available are there to provide alternatives. This suggests that, if one communication mean is not available, one can simply use another and it works! But this is not true, every communication technique is designed for certain applications and are not always interchangeable. In this regard, the authors should present the existing communications, their applications and situations when one solution can be used instead of another.

R = Paragraphs of section 2.1 were updated to better explain the concept of communication technology selection.

  1. Figure 4 should present more accurate VRU categories. For example, you have "road user outside the vehicle", but this also includes "other road users", "special road user", even "user of transport devices", because all of them are outside a vehicle.

R = The classification of pedestrian types presented in section 2.2 was updated. The new design of figure 4 was made with the representation of the new classification.

  1. In chapter 3.2.1 are presented several communications technologies inside vehicles. A table that compare them would be useful, including main parameters and the applications.

R = Table 1 was added on page 8 to summarize the transmission media used in the cars for the interconnection of their devices and controllers.

  1. Please, document some more the LASER technology and decide if it uses  pulsed laser, or laser beams. Or if it is the same thing and it doesn't matter.

R = The error in the handling of different terminology was corrected, leaving only the term laser light pulse. In addition, a paragraph was added to complement the information on LiDAR.

  1. Also, please document some more the video camera technology. First of all, you say that infrared camera can use wavelength down to 1nm. But this is not infrared, it's even beyond ultraviolet and towards X-rays! Second of all, combing the information presented, it results that video cameras can only work with GMLS. Are you sure this is the only solution? What about cameras with video processing units integrated. Do they require the same (huge) amount of communication bandwidth?

R = We extend the explanation of video camera technology in section 3.2

  1. Please, present and detail the prediction equation and the update equation you are referring to in the text.

R = A paragraph has been added to explain the equations. In addition, the corresponding equations were included in the text.

  1. All the technologies presented in chapter 3.2 are from the vehicle point of view. Aren't any solutions from the VRUs point of view? Because interaction means "reciprocal action" and what you present do not suggest reciprocity.

R = A paragraph (has been added to describe how the cell phone is visualized as the tool with which VRUs can interact with CAVs.

  1. In chapter 4 you introduced some solutions to interact with VRUs, but also from the vehicle point of view. There are more solutions out there, such as V2P, V2B and so on that you don't mention anything about.

R = A paragraph was added to section 4 describing how the V2P communication mode is being used to establish pedestrian-to-car communication.

  1. In the end, I must emphasize again that the whole paper should be reconsidered and maybe a native English speaker should be involved. There are so many sentences that don't make sense, that is difficult for me to list them all.

R = As suggested, readability of the paper was improved.

  1. And all the abbreviations should be explained in the text.

R = All abbreviations were explained in the text.

Round 2

Reviewer 1 Report

  Thank you for your answers. All the modifications in the manuscript meet the requirements.

Author Response

Thank you very much for your kind comments and valuable revisions.

Reviewer 2 Report

Almost all figures in the paper have very poor quality and should be enhanced.

Figure 3 is still not correlated with the text, because the blocks presented in the figure should have the same name in the text. It's almost impossible to know for sure, due to very poor image quality, but I believe that block 3 is named "Motion and car control" in the figure, and "Motion and vehicle control" in the text.

You have eliminated row numbers, so it's more difficult now to identify the exact spot of an issue, but in the end of page 5 you present some signal propagation problems. I think this is an important aspect that should be further documented. The key element for a CAV is the ability to properly communicate with other entities (as you said yourselves), at high speeds and with a very good success rate. You should properly document this: what tests have been made? What are the limitations of different technologies (maybe speed limit, or bandwidth limitation)? And, most important, how can this be tackled? What are the solutions to improve connectivity?

You should provide the sources for the information provided in chapter 3.2.1. For example, in Table 1 you say that LVDS (the acronym is not explained...) can communicate up to 40m. But there are studies (like this one: https://www.ti.com/lit/wp/snla204/snla204.pdf?ts=1654494115113&ref_url=https%253A%252F%252F) that say the typical distance is 15-20m. 

I still consider video cameras should be better detailed. There are many solutions with integrated video analysis that reduce the amount of data to be transferred at 1Mbps or even less. So, it's a mistake to present in Table 2 the camera as an equipment that requires a huge amount of data to be transferred. Even without integrated video processing units, there are 12Mpx cameras that transfer video streams at 3-4 Mbps. So what camera do you have in mind that will generate 3500Mbps data stream?

Regarding the VRU's perspective in chapter 3.2: I didn't find anything new compared  to the previous version. You have added something in chapter 4, but:

1. you didn't explain the technology: why do you consider it's important to detail in chapter 3 the technologies used for cars, but you disregard details about technologies that can be used by other traffic participants? Are they not equally important in the traffic system?

2. you have a limited view from the VRU perspective, although the paper should focus mainly on them (as the title suggests): there is a trend now to use bikes and scooters, decreasing the usage of cars. Why don't you also consider these categories? There are VRU among them. You present mainly cars and a little bit the pedestrians, and I think this is wrong.

The chapter numbers are not in order! E.g. after 3.2.1 you have 3.1.2!

There are a lot of extra spaces in the text; it's not clear if it's something temporary for the reviewers to spot quicker the changes made, but the paper should be presented as it would be published, so I consider this to be a mistake that should be corrected.

English language is still a problem and I emphasize again the necessity for a native English speaker to be involved. There are grammar mistakes, words misspelled, etc.

Author Response

Thank you for your revisions. Please see attached document.
